# Bacterial, Archaeal, and Eukaryote Diversity in Planktonic and Sessile Communities Inside an Abandoned and Flooded Iron Mine (Quebec, Canada)

Elise Lhoste [1,2], Francis Comte [1], Kevin Brown [3], Alain Delisle [3], David Jaclin [4], Violaine Ponsin [5], Maikel Rosabal [1,2] and Cassandre Sara Lazar [1,2,*]

1    Department of Biological Sciences, University of Québec at Montréal (UQAM), Montréal, QC H2X 1Y4, Canada
2    Interuniversity Research Group in Limnology/Groupe de Recherche Interuniversitaire en Limnologie (GRIL), C.P. 6128, Succ. Centre Ville, Montréal, QC H3C 3J7, Canada
3    PTO Exploration, Ottawa, ON K1N 5H3, Canada
4    Faculty of Social Sciences, University of Ottawa, Ottawa, ON K1N 9A7, Canada
5    Department of Earth and Atmospheric Sciences, and GEOTOP, University of Quebec at Montreal, C.P. 8888, Succ. Centre-Ville, Montréal, QC H3C 3P8, Canada
*    Correspondence: lazar.cassandre@uqam.ca; Tel.: +1-(514)-987-3000 (ext. 3963)

**Abstract:** Abandoned and flooded ore mines are examples of hostile environments (cold, dark, oligotrophic, trace metal) with a potential vast diversity of microbial communities rarely characterized. This study aimed to understand the effects of depth, the source of water (surface or groundwater), and abiotic factors on the communities present in the old Forsyth iron mine in Quebec (Canada). Water and biofilm samples from the mine were sampled by a team of technical divers who followed a depth gradient (0 to 183 m deep) to study the planktonic and sessile communities' diversity and structure. We used 16S/18S rRNA amplicon to characterize the taxonomic diversity of Bacteria, Archaea, and Eukaryotes. Our results show that depth was not a significant factor explaining the difference in community composition observed, but lifestyle (planktonic/sessile) was. We discovered a vast diversity of microbial taxa, with taxa involved in carbon- and sulfur-cycling. Sessile communities seem to be centered on $C_1$-cycling with fungi and heterotrophs likely adapted to heavy-metal stress. Planktonic communities were dominated by ultra-small archaeal and bacterial taxa, highlighting harsh conditions in the mine waters. Microbial source tracking indicated sources of communities from surface to deeper layers and vice versa, suggesting the dispersion of organisms in the mine, although water connectivity remains unknown.

**Keywords:** iron mine; aquatic subsurface ecosystem; genomics; bacteria; archaea; eucaryote; planktonic; sessile





## 1. Introduction

As a result of Canada's long history of mining, the country possesses many abandoned ore mines, estimated at ca. 10,000 sites [1]. Because mine closure and rehabilitation were not regulated before 1991 [2], when mining activities would cease, sites would simply be abandoned without environmental restoration. Added to the fact that mining exploitation leads to the destruction of natural habitats and the disturbance of underground ecosystems, abandoned mines are also an important source of environmental problems [3]. Previous studies have shown the negative impact of mining residue on the fauna and flora in surrounding soils [4–6], surface waters [7], or groundwater [8]. Although mining closure and restoration are now part of Canadian jurisdiction, in 2020, the province of Quebec possessed 223 abandoned mining sites with no planned restoration [2].

When mines are in activity, digging in subsurface layers leads to flooding of the produced underground tunnels by groundwater, which needs to be pumped. As a result,

when an owner ceases its mining operation, groundwater rushes into the abandoned mines' tunnels and shafts to restore the natural level of surface waters [9], thus creating a novel artificial aquatic subsurface ecosystem over time. Water circulating in the mine will come into contact and dissolve minerals, which may modify the water quality of the mine [10]. Therefore, microbial communities already present in the mine soil or on the rock walls, as well as in the groundwater, have to adapt to this novel artificial habitat with new environmental conditions that can be qualified as hostile [11]. Indeed, away from the surface waters, oxygen decreases gradually with the increases of with pressure/depth, as well as electron donors and acceptors, nutrients, temperature, and luminosity. Therefore, depth (from surface to subsurface water) will potentially act as the main gradient leading to shifts in microbial communities inside flooded ore mines [12]. This interaction between surface and subsurface water inside ore mines can be studied using water stable isotopes ($^{18}O$ and $^{2}H$). Because microorganisms are abundant and ubiquitous, they are found in even the most extreme ecosystems [13], reflecting the environmental conditions of their surroundings. Thus, microorganisms isolated from these habitats could be tools to describe and understand hostile environment, such as abandoned and flooded ore mines. Furthermore, because these artificial aquatic habitats are unique and isolated, and many have been stablished for decades, they could potentially hide a plethora of microbes adapted to these conditions, unknown in our current databases.

Currently, studies of microbial communities living in flooded ore mines are still very limited. Most studies focus on microbial diversity in acid drainage, or caves [13,14]. Recently, some submerged ore mines have been studied, like gold, uranium, vanadium, and copper mines [12,15–17]. In gold and vanadium mines, a vast relative abundance of unclassified bacteria was observed [13], showing many non-culturable microorganisms in these mines with possible biotechnological interests. Moreover, there are only a few studies encompassing microbial diversity living in both biofilms (sessile) and in water (planktonic), as is the case in most aquatic ecosystems [18–21]. Living in biofilms brings many benefits to members, allowing them to cope with hostile conditions and protect their cells within the biofilms [22]. Living freely in the water, however, allows cells to actively search for nutrients or less hostile conditions [23]. Transition from one lifestyle to another typically depends on environmental conditions. Overall, our knowledge about the communities living inside flooded mines is lacking, mainly explained by the extreme difficulty in accessing these environments.

Therefore, in this study, we analyzed microbial communities living inside the old Forsyth iron mine in the province of Quebec, Canada, where the temperature reaches 5.5 °C and visibility is limited across the depth gradient [24]. We targeted all domains of microbial life (Archaea, Bacteria, and Eukaryotes) to gain understanding of the microbial communities as a whole using 16S/18S rRNA Illumina gene sequencing. The goals of this study were to (1) compare the diversity of sessile and planktonic communities across the depth gradient; (2) investigate the surface and groundwater connectivity through the depth gradient of the mine; and (3) examine the influence of environmental factors on the microbial community structure. Here, we shed light on an iron mine microbiome and provide a better understanding of the diversity and the connectivity of surface-groundwater microorganisms in relation with environmental factors. To the best of our knowledge, this genomic study is one of the first on abandoned and flooded mines in Canada, highlighting a huge microbial diversity. This study will improve our understanding of these microbial communities, allowing us to develop strategies for the bioremediation and rehabilitation of mining sites and polluted aquatic ecosystems in general.

## 2. Materials and Methods

### 2.1. Site Description and Sample Collection

The Forsyth mine (MF) is an old iron mine located in the Gatineau Park in Quebec (Figure 1) part of the National Capital Commission (NCC). It was discovered in 1801 and mining activities started in 1855 until 1918, when all operations ceased. For this study,

samples were collected manually in August and September 2019 by the PTO Exploration team, specialized in technical diving in extreme conditions. Nine sampling depths (0, 6, 12, 18, 24, 27, 61, 91, and 183 m) were selected to cover the mine excavated tunnels and shafts (Figure 1). From depths 0 to 24 m, water samples were taken in two 1 L polyethylene bottles, previously sterilized (Nalgene®, Rochester, NY, USA) for microbial identification and in three 50 mL syringes for physicochemical analysis. Biofilm samples were taken in three 50 mL separate syringes from wood, rock, and metal substrates from 0 to 24 m deep and at 27, 61, 91, and 183 m deep in the shaft. Sampling at 183 m was done during one individual dive. This constitutes the deepest known sample collected by a diver in a cold water mine. The diver involved in this feat reports that water became extremely clear—compared to the very turbid water in the tunnel and upper levels in the shaft—after 91 m [24]. All samples were stored at 4 °C in the dark during transport. For water samples, filtration for microbial analyses was carried out using a 0.2 µm polyethersulfone filter (Sartorius®, Midisart, Germany) on the same day as sampling. Filters and biofilms were stored at −80 °C until further processing.

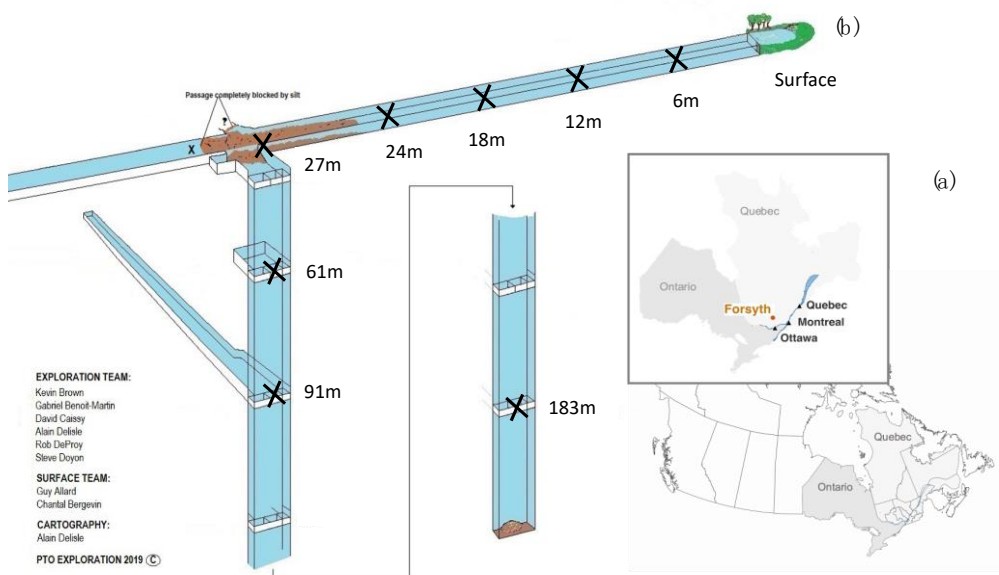

**Figure 1.** (**a**) Localization map of the Forsyth mine in Quebec, Canada (QGIS), (**b**) Forsyth mine cartography. X represent sampling localization.

## 2.2. Physicochemical Analyses

Water samples were filtered on 0.45 µm polyethersulfone filters (Sarstedt®, Numbrecht, Germany) for dissolved organic (DOC) and inorganic (DIC) carbon, nitrite, and nitrate analyses. Carbons were analyzed with an OI Analytical Aurora 1030W TOC Analyzer (College Station, TX, USA) using a persulfate oxidation method at the GRIL-UQAM (Groupe de Recherche Interuniversitaire en Limnologie) analytical laboratory. Nitrite and nitrate were analyzed with a continuous flow analyzer (OI Analytical Flow Solution 3100 ©, College Station, TX, USA) using an alkaline persulfate digestion method, coupled with a cadmium reactor, following a standard protocol [25]. Samples were analyzed at the GRIL-UQAM analytical laboratory. For ammonium and ammoniac analyses, water samples were filtered on 0.20 µm polyethersulfone filters (Sarstedt®, Numbrecht, Germany) and analyzed with a Flow Solution 3100 autosampler using a chloramine reaction with salicylate to form indophenol blue dye (EPA Method 350.1).

Anions ($F^-$, $Cl^-$, $SO_4^{2-}$) were analyzed with ion chromatography (Dionex aquion, Thermo Scientific, Waltham, MA, USA) at the Centre de recherche sur la dynamique du système Terre (GEOTOP)-UQAM analytical laboratory. Water samples ($n$ = 21) were analyzed for δ2H and δ18O at the stable isotope's laboratory of the GEOTOP-UQAM. Measurements were made using a dual inlet Micromass Isoprime™ isotope ratio mass spec-

trometer coupled to an Aquaprep™ system for δ18O and using Isotopic Water Analyzer for δ2H. For oxygen and hydrogen isotopic analyses, 1 mL of water was pipetted in a 2 mL vial and closed with a septum cap. The samples were analyzed with an LGR (Los Gatos Research) model T-LWIA-45-EP Off-Axis Integrated Cavity Output Spectroscopy (OA-ICOS). Each sample was injected (1 microliter) and measured 10 times. The isotopic compositions of samples were corrected using three internal reference waters ($\delta^{18}O$ = 0.23 ± 0.06‰, −13.74 ± 0.07‰ and −20.35 ± 0.10‰; $\delta^2H$ = 1.28 ± 0.27‰, −98.89 ± 1.12‰, and −155.66 ± 0.69‰; $\delta^{17}O$ = 0.03 ± 0.04‰, −7.32 ± 0.06‰, and −10.80 ± 0.06‰) calibrated on the Vienna Standard Mean Ocean Water–Standard Light Antarctic Precipitation (VSMOW-SLAP) scale [26]. The overall analytical uncertainty (1s) is better than ±0.1‰ for $\delta^{18}O$, ±1.0‰ for $\delta^2H$, and ± 0.1‰ for $\delta^{18}O$. Values are reported in per mil units (‰) against the Vienna Standard Mean Ocean Water standard (VSMOW).

### 2.3. DNA Extraction, 16S and 18S rRNA Gene Amplification, and Sequencing

DNA was extracted from the filters using the DNeasy PowerWater kit (Qiagen, Hilden, Germany) according to the manufacturer's instructions. The bacterial V3–V4 region of the 16S rRNA gene was amplified using the primer pair B341F (5′-CCTACGGGAGGCAGCAG-3′) and B785R (5′-GACTACHVGGGTATCTAATCC-3′) [27]. The archaeal V3–V5 region of the 16S rRNA gene was amplified using the primer pair A340F (5′-CCCTACGGGGYGCAS-CAG-3′) and A915R (5′-GTGCTCCCCCGCCAATTCCT-3′) [28]. The Eukaryotic V5 region of the 18S rRNA was amplified using the primer pair E960F (5′-GGCTTAATTTGACTCAA-CRCG-3′) [29] and NSR1438R (5′-GGGCATCACAGACCTGTTAT-3′) [30]. PCR reactions (Supplementary Material Table S1) were performed using the Phusion Hot Start II polymerase (Thermo Scientific™), following the manufacturer's instructions. Sequencing for each domain was carried with Illumina Miseq using MiSeq Reagent Kit v.3 (600 cycles, illumina) at the CERMO-FC genomic platform (Centre excellence en recherche sur les maladies orphelines—fondation Courtois). Negative controls for DNA extraction kit and PCR amplification were sequenced for all three domains. The raw reads have been deposited into the National Center for Biotechnology Information (NCBI) under the PRJNA916235 accession number.

### 2.4. Sequence Data Analysis

To identify amplicon sequence variants (ASVs), 16S/18S rRNA gene sequences were filtered, processed, and analyzed using a modified DADA2 pipeline (v.1.24.0) [31] in R v4.2.2, to manage the quality of forward and reverse reads. For bacterial sequences, forwards and reverse reads were truncated at positions 250 and 200, respectively. For eukaryote sequences, forwards and reverse reads were truncated at positions 260 and 200, respectively. For archaeal sequences, because of the low quality of the reverse reads, only forward reads were kept and truncated at position 260. Taxonomy was assigned with the DADA2 package using the assign taxonomy function and the SILVA SSU database (v.138.1 for bacterial and archaeal sequences, v.132 for eukaryotes sequences). Non-classified archaeal and eukaryotic sequences were further classified using a personal database for the Archaea construct based on Liu et al. (2018) [32] and Zhou et al. (2018) [33] and the PR2 SSU database (v.4.14.0) for the Eukaryotes. ASVs in negative controls (extraction kits and PCR) were removed from all output sequences, with the decontam R package (v.1.18.0, [34]) using the is Contaminant function. Then, bacterial, archaeal, and eukaryotes community tables were rarified to 1500, 20,000, and 5700 ASV, respectively. Only samples containing ASVs assigned to the three domains were kept. For the Eukaryotes, sequences affiliated with vertebrate eukaryotes taxa were removed to retain only microbial eukaryotes.

### 2.5. Statistical Analysis

Statistical analyses were performed with Rstudio Server 2022.07.2, and for all tests, differences were considered statistically significant if *p*-value < 0.05. To analyze the connectivity of the mine with the bacterial, archaeal, or eukaryotes communities across the depth

gradient and the relationship between sessile and planktonic communities, we used a series of diversity indices related to taxonomy, including the relative abundance, ASV richness and the Shannon index for each depth, lifestyle community (sessile or planktonic), and substrate type for the sessile communities (metal, rock, or wood), using the Phyloseq package (v.1.40.0, [35]). We used the non-parametric Kruskal-Wallis test to evaluate the difference of bacterial, archaeal, or eukaryotes $\alpha$-diversity between depth and substrate type, followed by a post-hoc Dunn test to performs multiple pairwise comparison, using the ggpubr package (v.0.5.0) and the compare_means function. The difference in bacterial, archaeal, or eukaryotes $\alpha$-diversity between the sessile and planktonic communities was tested using a non-parametric Wilcoxon test, using the ggpubr package and the compare_means function. In addition, the relationship between abiotic factors and depth or $\alpha$-diversity were determined through Spearman correlation analyses using ggplot2 function (v.3.4.0).

To visualize dissimilarities of bacterial, archaeal, or eukaryotes communities across the depth gradient and between lifestyles (sessile and planktonic), a Principal Coordinates Analysis (PCoA) was performed on the rarefied relative abundance data using a Bray-Curtis dissimilarity matrix with the Vegan package and the vegdist funstion (v.2.6.4, [36]). Variation partitioning analysis was done based on PERMANOVA with significance tested using the default 999 permutations, using the Vegan package (v.2.6.4, [36]) and the adonis2 function. Permutation homogeneity tests to assess differences in $\alpha$-diversity between lifestyle, substrate types, and across the depth gradient were also performed using the Vegan package (v.2.6.4, [36]) and the betadisper and permutest functions.

To detect microbial taxa explaining the differences between the two lifestyles, linear discriminant analysis effect size (LEfSe) "https://huttenhower.sph.harvard.edu/galaxy (accessed on 10 December 2022)" [37] was applied at the genus level on rarefied data (LDA threshold > 2, $p < 0.05$). Then, to characterize the microbial communities interactions between the surface water and the groundwater, we performed fast expectation-maximization microbial source tracking (FEAST) using the Feast package (v.0.1.0, [38]). This method estimated the contribution (proportion) of communities of one depth as a source to another as a sink [39].

## 3. Results

### 3.1. Environmental Variables along the Depth Gradient

The mine water was characterized by a neutral pH and a temperature of about 5 °C. In addition, the environmental parameters ($SO_4^{2-}$, $Cl^-$, $F^-$, DIC, DOC) followed the depth profile with all the parameters except DOC increasing with depth (Supplemental Material Figure S1). The increase of $SO_4^{2-}$ and $Cl^-$ concentrations were significantly correlated with the depth gradient (Spearman correlation, $p = 0.017$; Supplemental Material Table S2).

Surface water displayed $\delta^{18}O$ values between $-12.28$ and $-11.93‰$ and $\delta^2H$ values between $-83.16$ and $-81.4‰$. Mine water displayed $\delta^{18}O$ values between $-12.35$ and $-11.36‰$, and $\delta^2H$ values between $-84.10‰$ and $-76.77‰$. The low values of $\delta^{18}O$ and $\delta^2H$ reflects the major influence of snow melt and the recharge of the mine water. The mine water data was very scattered and deviated from the GMWL, and most of it was located above the GMWL and along the line with a slope of 3.3 ($\delta^2H = 3.3842\,\delta^{18}O - 41.963$; $R^2 = 0.1741$), indicating that the water was affected by evaporation (Supplemental Material Figure S2).

### 3.2. Alpha Diversity and Correlation with Environmental Variables

Alpha diversity indices of Bacteria, Archaea, and Eukaryotes in the mine were measured by the ASV richness and the Shannon index. The Shannon index value and ASV richness varied for the Bacteria, Archaea, and Eukaryote domains (Supplemental Material Figure S3). For the three domains, differences in alpha diversity indices were not significantly related to the depth. However, the Shannon index and ASV richness of the eukaryote community were significantly different between lifestyles ($p = 0.00047$, $p < 0.00001$ respectively, Supplemental Material Figures S4 and S5). The sessile communities had Shannon

index and ASV richness significantly higher than the planktonic communities, particularly between rock-surface and water populations ($p = 0.008$, $p = 0.0004$). In contrast, alpha diversity indices of bacterial and archaeal communities were not significantly correlated to lifestyles.

In addition, Spearman correlations between alpha diversity indices and depth or all physicochemical parameters were not significant for archaeal and eukaryota communities, as shown in Table S3, but a significant negative correlation ($r = -0.713$, $p = 0.014$, Supplemental Material Table S3) was observed between depth and the ASV richness of bacteria communities.

### 3.3. Microbial Community Structure

Proteobacteria were the most abundant bacterial phyla found in the mine (64.2%), followed by the Nitrospirota (19.6%) (Figure 2a), especially in the sessile communities (Supplemental Material Figure S6a). Nanoarchaeota (33.8%) and Crenarchaeota (31.7%) were the two archaeal phyla found at the same relative proportion (Figure 2b, Supplemental Material Figure S6b). Eukaryote phyla were richer with the Opisthokonta as the most abundant (20.5%), particularly found in the sessile communities, followed by the Ciliophora (11.9%) found in all samples and the Arthropoda (11.4%) found in the surface and 5 m water samples (Figure 2c, Supplemental Material Figure S6c).

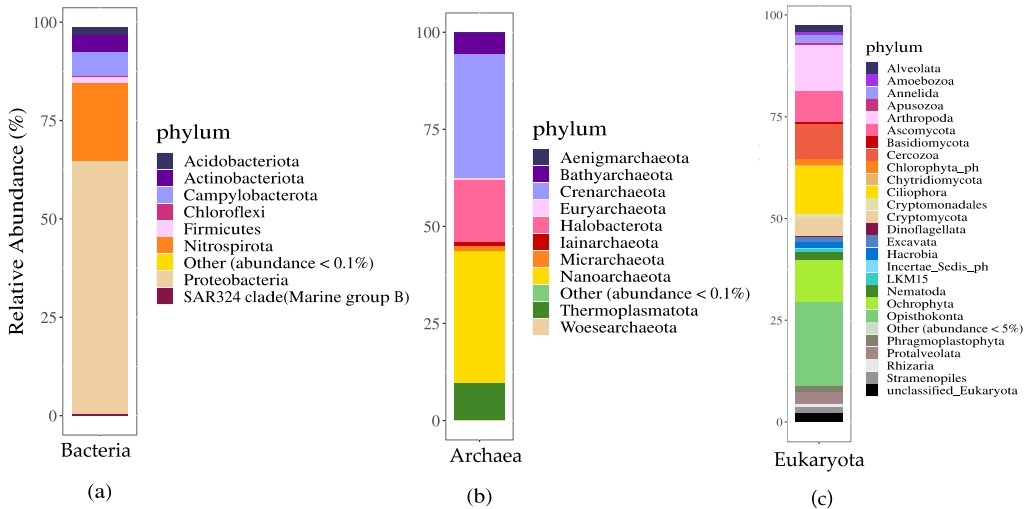

**Figure 2.** Relative abundance (% of total reads) of bacterial (**a**), archaeal (**b**), and eukaryote (**c**) taxa at the phylum level. For each domain, means of all samples are presented, with *n* = 37, *n* = 34 and *n* = 37, for Bacteria, Archaea and Eukaryotes, respectively.

The principal coordinates analysis based on Bray-Curtis dissimilarities revealed a separation following lifestyle along the first axis, which explained 19.2%, 23.5%, and 17.1% of the variation, for the bacterial, archaeal, and eukaryota communities, respectively (Figure 3). PERMANOVA confirmed that the lifestyle adopted by the microbial communities was a significant variable, explaining 14.6%, 17.4%, and 11.4% of the variance for each domain (Table 1). Depth was not an environmental factor significantly explaining variance for either domain (Supplementary Materiel Table S4). However, for the sessile and planktonic bacterial communities, the PCoA plot showed a separation between surface communities (0 to 12 m for planktonic communities and 0 to 24 m for sessile communities) and deep communities (24 to 183 m). For eukaryotes, only planktonic communities seem to be different between surface (0 to 12 m) and deep (18 to 24 m) water.

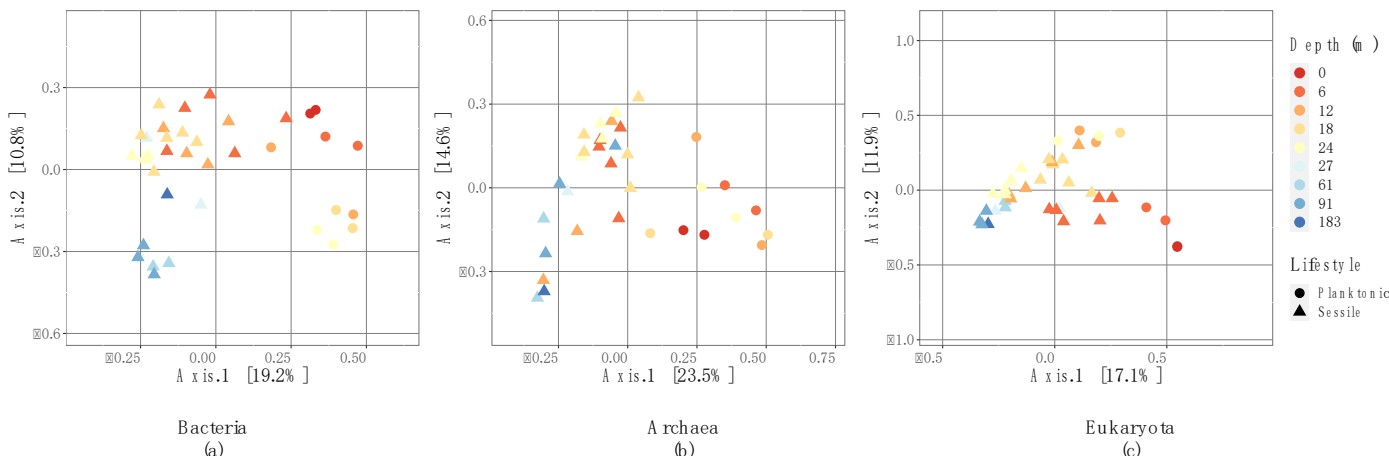

**Figure 3.** Principal coordinates analysis (PCoA) ordination of bacterial (**a**), archaeal (**b**), and eukaryote (**c**) community variance based on a Bray-Curtis dissimilarity matrix. The dotted lines indicate 95% confidence interval groups by lifestyle.

**Table 1.** Variation in bacterial, archaeal and eukaryote community composition explained by depth, lifestyle (planktonic or sessile) or substrates (water, wood, rock, metal), tested using PERMANOVA.

| | **Bacteria** | | | | | **Archaea** | | | | | **Eukaryota** | | | | |
|---|---|---|---|---|---|---|---|---|---|---|---|---|---|---|---|
| | dF | Sum of Sqs | $R^2$ | F | Pr (>F) | dF | Sum of Sqs | $R^2$ | F | Pr (>F) | dF | Sum of Sqs | $R^2$ | F | Pr (>F) |
| Depth | 1 | 0.959 | 0.08 | 3.812 | 0.001 | 1 | 0.794 | 0.094 | 4.292 | 0.001 | 1 | 1.23 | 0.091 | 4.469 | 0.001 |
| Lifestyle | 1 | 1.755 | 0.146 | 6.98 | 0.001 | 1 | 1.464 | 0.174 | 7.919 | 0.001 | 1 | 1.337 | 0.099 | 4.859 | 0.001 |
| Residual | 33 | 8.297 | 0.692 | | | 30 | 5.548 | 0.659 | | | 36 | 9.905 | 0.73 | | |
| Depth | 1 | 0.959 | 0.08 | 4.187 | 0.001 | 1 | 0.794 | 0.094 | 4.703 | 0.001 | 1 | 1.23 | 0.091 | 4.434 | 0.001 |
| Substrate | 4 | 2.894 | 0.241 | 3.161 | 0.001 | 4 | 2.424 | 2.288 | 3.591 | 0.001 | 4 | 2.31 | 0.17 | 2.082 | 0.001 |
| Residual | 27 | 0.515 | 0.515 | | | 24 | 4.051 | 0.481 | | | 30 | 8.321 | 0.613 | | |
| Total | 36 | 11.996 | 1 | | | 33 | 8.418 | 1 | | | 39 | 13.565 | 1 | | |

*3.4. Discriminative Microorganism's Taxa According to Lifestyle (LEfSe)*

LEfSe between the two lifestyles, which were significant variables explaining microbial variance, showed six bacterial genera significantly higher in the planktonic communities, but no genus was significant in the sessile communities (Figure 4a). These included unclassified (unc.) Sulfurospirillaceae, Alcaligenaceae, and Micropepsaceae. For the Archaea, one genus was significantly higher in the sessile communities (unc. Bathyarchaeota subgroup 22) while 17 were significantly higher in the planktonic communities (Figure 4b). For the Eukaryotes, 7 genera were significantly higher in the sessile communities while 10 were higher in the planktonic communities, all belonging to different phyla (Figure 4c).

*3.5. Microbial Source Tracking: Surface and Groundwater Contributions*

We used fast expectation-maximization for microbial source tracking (FEAST) to infer on potential flows of water from the surface to the deeper tunnels, or vice versa, which could transfer microbial cells from one depth to the other. Going in either direction, we analyzed separately the planktonic communities since it is more likely that these will move or be transported (Figure 5). For each depth, we used the previous depths as potential sources. However, we also analyzed the planktonic and sessile communities together since there are constant exchanges between both types of lifestyles (Figure 6). Here, for each depth, we used the planktonic and sessile communities from the previous depths as potential sources.

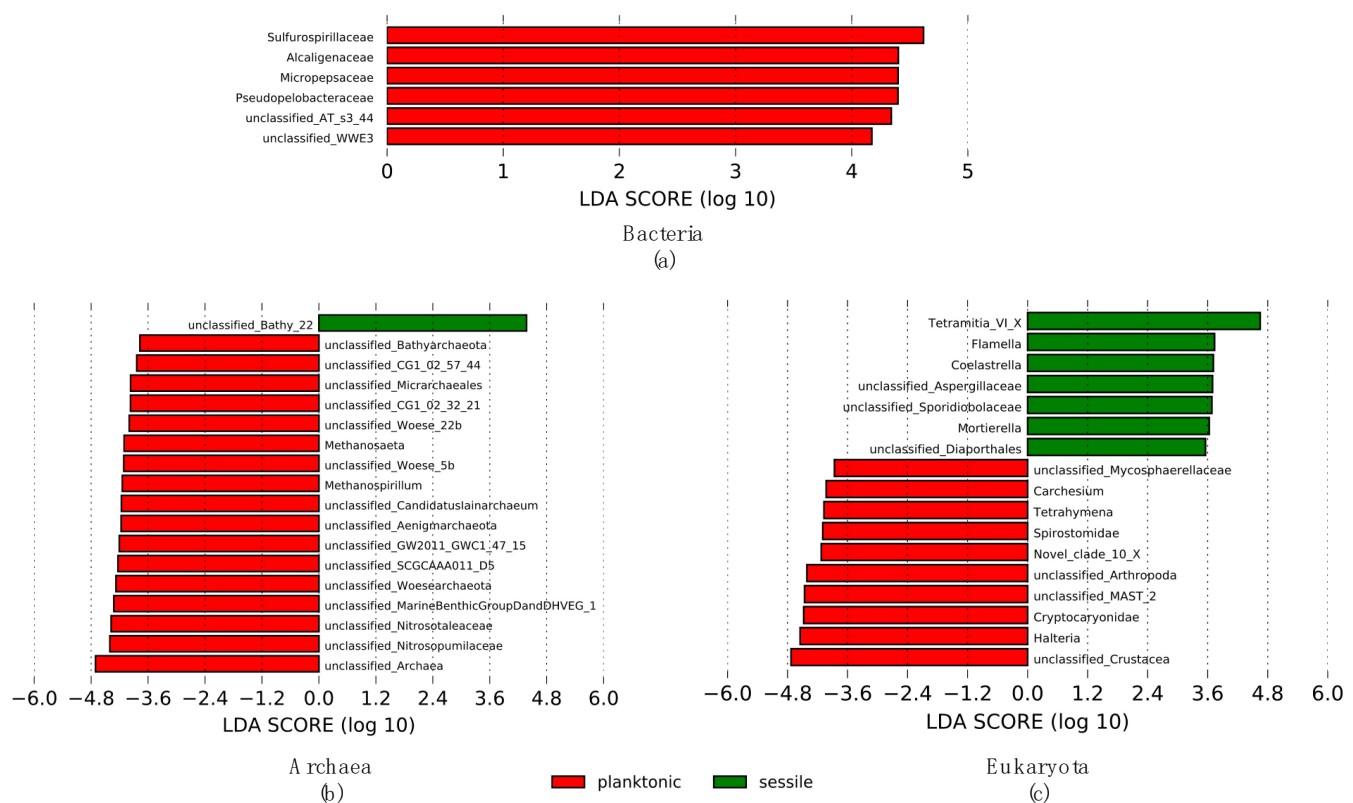

**Figure 4.** Discriminative features of bacterial (**a**), archaeal (**b**) and eukaryotal (**c**) taxa between planktonic and sessile communities at the genus level, using LEfSe analysis (log LDA threshold > 2, $p < 0.05$).

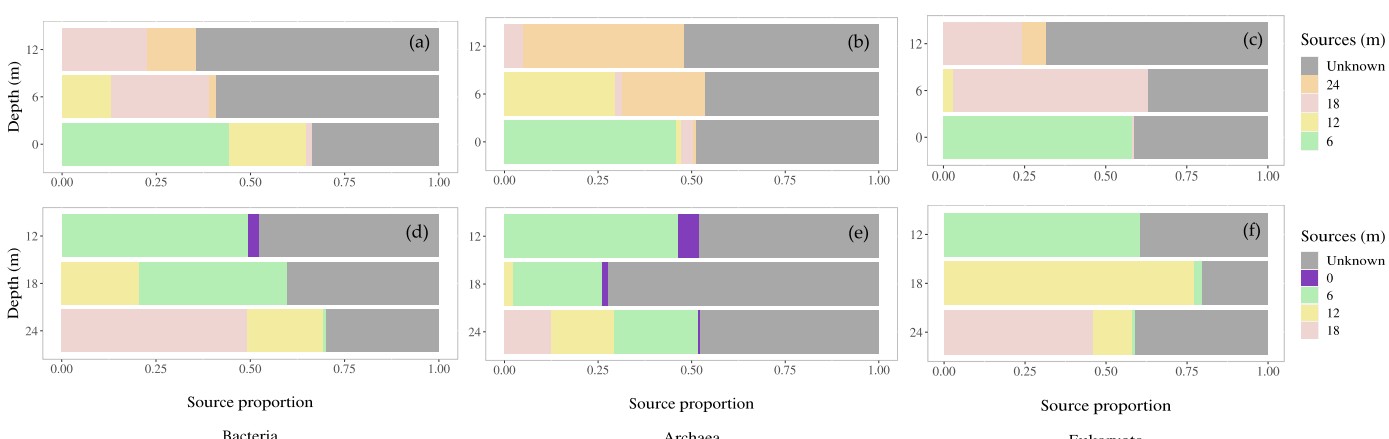

**Figure 5.** Microbial source tracking analyses estimating the contribution of planktonic bacterial (**a**), archaeal (**b**) and eukaryote (**c**) communities from the deeper (12 m) into the surficial layers or bacterial (**d**), archaeal (**e**), eukaryoteota (**f**) from surficial (12 m) into the deeper layer (24 m) communities, across the depth gradient (m). The unknown sources are indicated in grey. Each source correspond to a layer in the depth gradient (m).

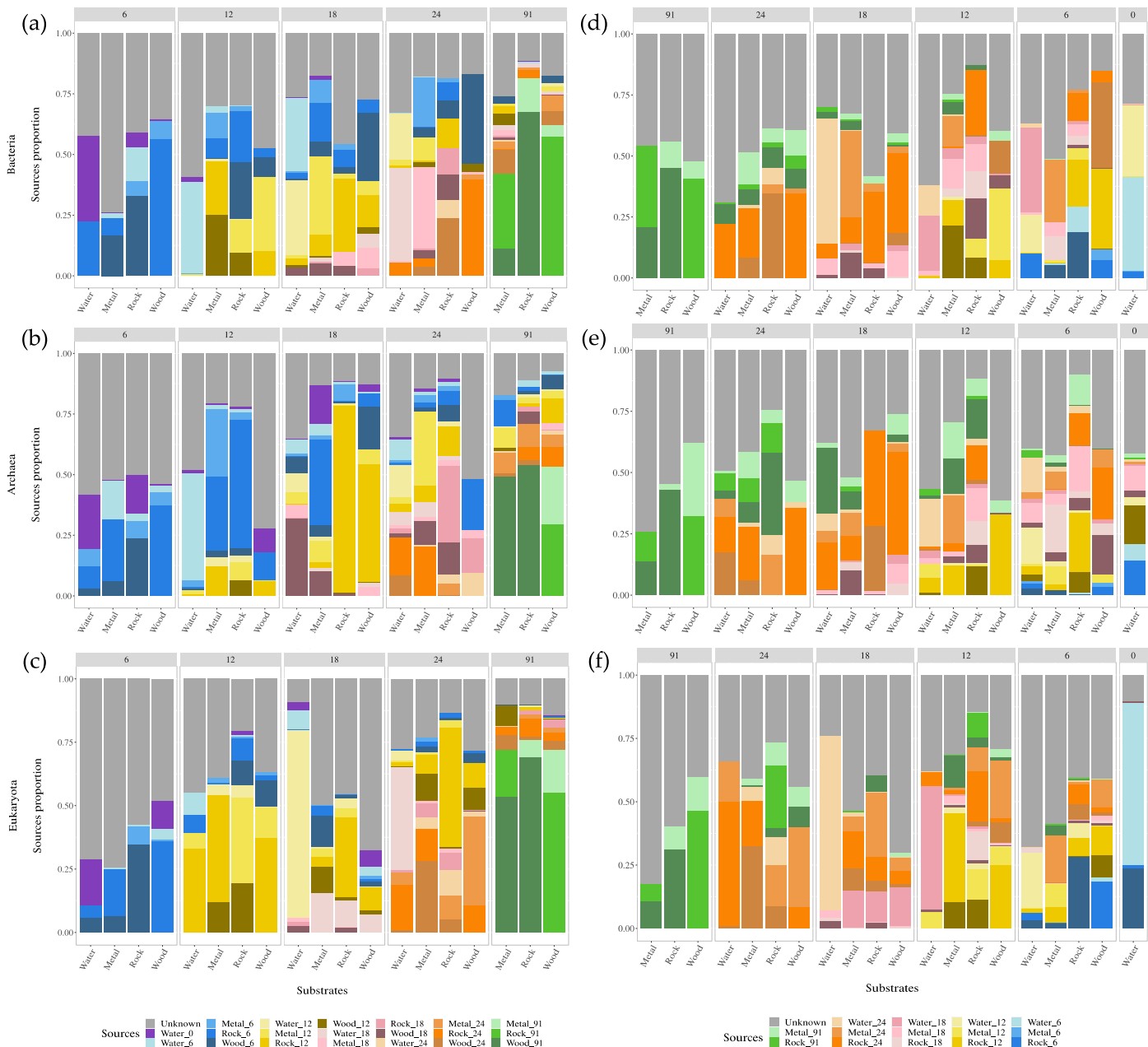

**Figure 6.** Microbial source tracking estimating the contribution of planktonic and sessile bacterial (**a**), archaeal (**b**) and eukaryote (**c**) communities from surficial into the deeper layers communities, or bacterial (**d**), archaea (**e**) and eukaryote (**f**) communities from the deeper into the surficial layers across the depth gradient and in the different substrates. Each color indicates a source corresponding to a substrate at a specific depth. Unknown sources are indicated in grey.

For the planktonic communities (when examined independently), the exchanges of eukaryote communities in the water column seemed less dynamic than the other domains (Figure 5c,f). However, unknown sources were more important for archaeal communities than bacterial and eukaryote communities (52.5%, 46%, and 41% respectively). From the surficial to deeper layers, archaeal communities in the surface water were found in all deeper layers (Figure 5b). From the deeper to the surficial layers, the sources of bacterial communities in the different layers were more variables than for the other domains. In addition, there was a weak contribution of the 24 m layer to the surficial layers (Figure 1), except for the archaeal communities (Figure 5e).

When we investigated both the planktonic and sessile communities together, the microbial source tracking analysis suggested transfers between both communities from the surficial to the deeper layers, and inversely (Figure 6). Overall, for the three domains, the contribution of the surficial communities to the deeper communities evolved in the same way along the depth gradient (Figure 6a–c). The surface planktonic communities contributed strongly to the formation of the planktonic communities 6 m deep and biofilms from different substrates enriched each other. In addition, the surface water played a larger contribution to the archaeal communities at 12 m deep than the other domains, mostly composed of sessile communities from the same depth (Figure 6a–c). At 18 m deep, the sources of the planktonic communities were diversified for the three domains, especially for the archaea, with a dominance of the 12 m wood communities. Proportions of surface water communities were again present at 18 m deep, especially for the archaeal communities in sessile communities using the metal substrate. At 24 m deep, the sources of the planktonic communities were diversified for the three domains, especially for the archaea. The contribution of surficial layers was lower for the eukaryote communities than for archaeal and bacterial communities, and the 18 m planktonic communities were the major contributors for the bacteria and eukaryote communities. Only archaeal communities showed a potential contribution of surface planktonic communities at this depth. Finally, at 91 m deep, the three domains had very low unknons sources. The surficial layers, especially 6 and 12 m, were bigger sources in the archaeal communities than the other domains.

From the deeper to surficial layers, the results showed that at 91 m, sessile communities' contribution was similar with different proportions (Figure 2d–f). At 24 m, the deeper layer communities were sources, except for eukaryote planktonic communities, which enriched rock and metal communities from 24 m deep. At 18 m deep, bacterial and eukaryote planktonic communities had a strong origin from planktonic communities at 24 m deep. However, archaeal communities from 18 m deep had important proportions of wood and rock communities from 91 and 24 m, respectively. In addition, for the three domains, the sources of communities present in different substrates were variables. Bacterial and eukaryote planktonic communities seemed less dynamic in terms of exchange compared to the archaeal planktonic communities. At 12 m deep, there was no contribution of communities from the 91 m layer for bacteria, and it was very low for archaea and eukaryotes. The planktonic communities from the surface were more diverse in the archaeal communities than the bacterial and eukaryote communities with a large contribution from the 6 m layers.

## 4. Discussion

Abandoned and flooded ore mines are generally considered as extreme environments because the availability of organic matter, oxygen, and light are limited. However, diverse microbial communities can colonize these habitats. Since the closure of the Forsyth mine, the inside tunnels and shafts are only accessible by a surface pond leading inside the mine entrance. Diving in extremely difficult conditions is the only way inside the mine, so any anthropogenic external microbial contamination would be very limited, metal-originated sources in the mine are also really limited due to its distance from industrial and urban centers.

### 4.1. Effect of Depth on Environmental Variables and Microbial Communities

The Forsyth mine has a large entrance, therefore conditions inside the mine can be influenced by temperature and nutrient inputs carried via surface water seeping in, leading to changes in food webs and ecosystem productivity [40]. The DOC concentrations support this assumption, since concentrations decreased with depth, suggesting a depletion of the surface DOC in the deeper water likely linked to microbial activities [41]. However, sulfate concentrations increased with depth, potentially explained by the presence of sulfate in groundwater flowing from deeper sources to the surface. Sulfate is important in biochemical cycles because some microbes oxidize organic matter using sulfate as the

terminal electron acceptor under anaerobic conditions, i.e., sulfate-reducing bacteria [41–43]. The variations in isotopic composition were within narrow limits (2‰ or 3‰ for $\delta^2$H and $\delta^{18}$O) result suggesting well-mixed underground systems or slow flow recharge pathways.

However, for the three microbial domains of life, no significant effect of depth as an environmental variable was observed either on alpha or beta diversity indices. Only bacterial ASV richness was negatively correlated to depth, suggesting a depletion of bacterial richness in deeper water layers of the mine, probably due to a decrease in available carbon, energy sources, and nutrients. These results are not consistent with previous studies indicating that changes in microbial diversity were primarily correlated with environmental conditions associated with depth [44]. Although depth doesn't explain the variation of microbial structure, the second axe of the bacterial PCoA plot separated planktonic communities in two clusters, surficial (6–24 m) and deeper layers (61–183 m), indicating a shift in community structure. In the case of the Forsyth mine, it is possible that depth does not reflect variations of abiotic parameters and therefore cannot be used as an ecological filter. It is also possible that the flow of water is limited within the mine and that most nutrients have been depleted over time, and physico-chemical conditions have stabilized over time.

Although depth was not a significant variable, the PCoA plots showed separation of the biofilm samples in the tunnel with the biofilm samples in the shaft deeper depths. The biofilms at 27 and 61 m were relatively high in unc. Methyloligellaceae, and the deepest 183 m sample was dominated by the SWB02 genus of the Hyphomonadaceae. The deeper biofilms saw a relative increase in unc. Methanomassillicoccales, Unc. Bathyarcheia, *Nitrosoarcheum*, candidatus (cand.) Methanoperedens, and methanogenic genera (*Methanosarcina*, *Methanosaeta*, *Methanoregula*, *Methanobacterium*), with *Nitrosoarcheum* not a dominant archaeon in the 183 m biofilm. *Gastritricha*, *Colpodella*, and unc. Cryptomycota were the dominant eukaryotes in the 183 m biofilm. The deepest biofilm sample seemed to be dominated by $C_1$-compound cycling as methane-producing archaea were detected, with the dominant one (unc. Methanomassillicoccales) using methylated compounds (e.g., methanol) [44,45]. Methane-oxidizers (unc. Methyloligellaceae and cand. Methanoperedens) were present, likely using the methane produced by the methanogens [46–48]. However, unc. Methyloligellaceae, also previously detected in an ore mine in Sweden [49], can also use methylated compounds as carbon source. SWB02 was identified in two previous studies of mine ecosystems [49–51] and was suggested to be a syntrophic partner establishing electron transfers with methanogens [52], probably using amino acids as carbon source [53]. Cryptomycota have been described as dominant fungi in aquatic systems and are probably saprophytic organisms [54]. Methylated compounds are formed during biological degradation and originate from natural vegetation [54,55], and although we do not know how much surface organic matter seeps all the way down to 183 m in the Forsyth mine, there is likely little fresh carbon remaining at 183 m deep. The diversity of microorganisms presents in the deepest sampled biofilm of the mine support this assumption, with metabolisms mainly based on $C_1$-compound recycling.

### 4.2. Sessile and Planktonic Microbial Communities: Differences between Lifestyles

In contrast to the depth variable, lifestyle had a significant effect on the structure of microbial communities for all three domains.

### 4.2.1. Sessile Microbial Communities in the Forsyth Mine

For the sessile communities, the LEfSE analyses for the bacterial domain showed no significant explanatory genus while one archaeal genus was significantly higher: Bathyarchaetoa subgroup 22. The Bathyarcheota phylum is widespread in anoxic environments found in sediment biofilms and plants with a wide range of metabolic capacity. They are involved in carbon, nitrogen, or sulfur cycles [33] and are able to degrade biomass using anaerobic pathways involved in carbohydrate metabolism and acid-amino fermentation [56]. Some biofilm members were also shown to be involved in humic acid and

lignin degradation [57]. Bathyarchaeota subgroup 22 is part of ancient lineages found and adapted to hydrothermal environments [58]. However, as suggested by Feng and colleagues [58], this ancient lineage is probably adapted to cold environment like the 5 °C water in the mine. Moreover, the studies of Compte-Port and colleagues [56,58] describe a potential syntrophic interaction between Bathyarcheota and Thermoplasmata involved in carbon mineralization, and our results go in this sense since both were present in the different biofilms of the Forsyth Mine.

Although the LEfSe analysis did not reveal discriminative taxa in the bacterial sessile community, the relative abundance of Nitrospirota was higher in the sessile community of the mine (27% in the planktonic, and 73% in the sessile community). Within this phylum, the sulfate-reducing genus *Thermodesulfovibiniona* was relatively higher than in the water samples, especially at 61 and 91 m deep. These sulfate-reducing bacteria have a gene set for dissimilatory sulfate, thiosulfate, or sulfite reduction with a limited range of electron donors, such as formate, pyruvate, and lactate [59–61]. Thermodesulfovibrio also have a vast metabolic potential possessing fermentative pathways, hydrogen production or consumption and dissimilatory sulfate reduction, as shown in estuary sediments study [62]. *Thermodesulfovibiniona* are widely widespread in hostile habitats such as anaerobic niches associated with hot springs [60], thermophilic methanogenic sludges [61] deep terrestrial sub-surface [63] or in deep granitic groundwater [64], indicating an important environmental role, especially in the sulfur biogeochemical cycle. The higher sulfate concentrations in the deeper layers of the Forsyth mine likely explain this higher relative abundance of *Thermodesulfovibiniona.*

Similarly, the Crenarcheota archaeal phylum was mainly present in the sessile communities, although no Crenarchaeotal genus was found to be a significant explanatory taxon. For a long time, this phylum was typically described in niches with extremely difficult conditions, like high temperature or low pH [65]. To date, members of the Crenarcheota phylum are ubiquitous and can be detected in various niches, extreme or not, as groundwater [66], anaerobic granular sludge [67], sediment from acid mine drainage sediments [68], or a gold mine [69], suggesting a wide range of metabolic capacities [67]. In addition, [70] describe the potential prevalence of this heterotrophic phylum in metal contamination of freshwater, especially linked to zinc contamination in anoxic lake sediment, suggesting that anaerobic Crenarchaeota respond differently to metal stress than aerobic Crenarchaeota [71]. The Forsyth mine is an old iron mine, characterized as an anoxic environment with a presence of various trace metals, which can explain the large presence of Crenarcheota phylum. The probable accumulation of metals in the biofilms could also explain its prevalence in the sessile community in the mine. A few methanogenic genera were also relatively more abundant in the biofilm samples. Methanogens are commonly described as key players in biofilms, like in wastewater treatment systems, when hydrogen or acetate is provided within the biofilms [72].

In contrast with the archaeal and bacterial sessile communities, our study showed that α-diversity was significantly higher in sessile compared to planktonic communities for the Eukaryote domain. Protists such as Colpodella, Amoebozoa, and Gastrotricha were detected in the mine. Here, the amoebae likely act as secondary grazers and predators of the prokaryotic organisms. Because research on eukaryotes is largely neglected, there is not much data about physiological properties or ecological roles of these organisms. Nevertheless, studies on Gastrotricha reveal a high density in aquatic habitats, with aerobic and anaerobic metabolic pathways and low trophic level, feeding on bacteria, small algae, or detritus [72–75]. The presence of nematode in the water mine potentially indicate connectivity with surface waters because they are typically found in aquatic and terrestrial surface habitats [76]. The LEfSe analysis also highlighted almost as many genera significantly higher in both communities. The Opisthokonta group (fungi/metazoa) was the most abundant (20.5%), especially in sessile communities in the deep-water layers. Three fungi genera were significantly higher in the sessile communities: unc. *Aspergillaceae* (Ascomycota), unc. *Sporidiobolales* (Basidiomycota), and *Mortierella* (Mucoromycota). Recently,

Held and colleagues [77] have described mycological diversity in an old iron mine and identified Ascomycota as dominant fungi followed by Basidiomycota and Mucoromycota and characterized wood substrates as a fungal hotspot compared to other substrate in the mine. Substrates (wood, metal, or rock) can provide a carbon and nutrient source for fungi unique to subterranean environments. Generally, fungi can breakdown complex plant-derived molecules to simpler compounds, which would then be available to the prokaryotic community in the biofilms, specifically *Thermodesulfovibiniona,* Bathyarchaeota, Crenarchaeota, or methanogenic Archaea.

The fact that little to no archaeal and bacterial genera were significantly different in sessile compared to planktonic communities suggests many shared genera between the biofilms and water niches, with a higher number of unique taxa in the water. The more hostile but fluctuating environmental conditions found in the water niche likely selected a unique community with taxa not found in the biofilms. In addition, a small portion of the planktonic populations have the physiological abilities to colonize solid surfaces and form biofilms, but a larger portion of sessile organisms can return to a planktonic form [78]. In fact, adhesion of planktonic cells to the surface is mostly driven by surface-exposed components like flagella or fimbriae while the biofilm maturation mainly depends to the formation of the extracellular matrix compounds, such as exopolysaccharides.

### 4.2.2. Planktonic Microbial Communities in the Forsyth Mine

For the Bacteria, six genera were significantly higher in the planktonic communities, which were not dominant genera in the overall bacterial community (relative abundance < 0.5%). Among these genera, we detected the CPR (Candidate phyla radiation) phylum, with the WWE3 taxon from the Patescibacteria super-phylum, an ultra-small bacterium, grouping cells, and genomes with a size smaller than typical bacterial cells [79]. This super-phylum is found in a wide range of groundwater environments [80], and members of this group live in symbiosis with bacteria or archaea because they are missing many essential biochemical pathways [78–81]. Genera belonging to the Proteobacteria phylum were also significantly higher, such as unc. *Alcalingenacea,* which includes many species with applications in biotechnology and specific metabolic activity such as carbon or nitrogen from antimicrobial treatment compounds as energy source or chromate reductase activity (e.g., wasterwater treatment [82]; bioremediation of chromium from a mine, [83]). Unc. *Sulfurospirillaceae* from the Campylobacterota phylum were also significantly higher in the planktonic community, usually found in various habitats, such as soil, groundwater, and especially in polluted soil or habitats enriched in sulfur compounds [84,85].

For the archaeal planktonic communities, Woesearchaeales (Nanoarcheota phylum) was a determinative taxon as well as *Methanosaeta*, mainly found in the surface pond community and unc. Micrarchaeales belonging to DPANN super-phylum. The Nanoarchetota phyla (DPANN super-phyla) was one of the major phyla in both the sessile and planktonic communities but was found at higher relative abundance in the water niche. This phylum is observed in many different habitats with a range of environmental conditions [32], such as groundwater [86], freshwater [87], or terrestrial [88] ecosystems. This phylum is characterized by ultra-small archaea with limited metabolic capacities and obligate symbionts with other archaea [89] or episymbiont with CPR bacteria [86], which are notably methanogenic microbes [90] and also significantly higher in the planktonic community. Other archaea with small genomes and limited metabolic capabilities were significant in the water niche (candidatus Iainarchaeum; [91]). DPANN Woesearchaeota contributed a large proportion of the archaeal community, notably in the anoxic and oligotrophic biotopes, suggesting an adaptation of DPANN archaea to constrained and oligotrophic systems. In addition, we can note the importance of unclassified Archaea in the planktonic communities, suggesting the presence of new species adapted to the distinctive conditions found in the Forsyth mine.

Ciliophora (11.9%) and Arthropoda (11.4%) were the most abundant phyla in the planktonic communities of surficial water layers. Planktonic ciliates are ubiquitous and cover many ecological niches, such as microplastic biofilm [92] or acid mine drainage [93]

and play an important role in nutrient recycling [94]. They contain organisms belonging to a wide range of trophic layers from heterotrophs (bacterivores, herbivores, omnivores, carnivores, etc.) to symbionts (commensals, parasites, mixotrophs) [94–97].

Overall, the sessile community in the Forsyth mine was mainly composed of fungi probably acting as the main decomposer in the biofilms, heterotrophic bacteria involved in the sulfur cycle, heterotrophic and methanogenic archaea likely benefitting from production of smaller carbon-based molecules within the biofilm. It is probable that all these microorganisms are adapted to heavy metal stress as these tend to accumulate within biofilms. However, this assumption needs to be verified in future studies. The planktonic community was dominated by ultra-small archaea and bacteria, probably explained by the oligotrophic conditions found in the mine water. The eukaryote populations are involved in nutrients cycling and seem to be prokaryote predators, adding another stress to these organisms. Bacterial metabolisms also seemed to be based on sulfur- as well as nitrogen-cycling.

### 4.3. Connectivity of Surface and Mine Water and Impact on Microbial Communities

To understand the structure of the microbial communities, we need to identify the protentional origins and sources of microbial cells [39]. In the case of the Forsyth mine, there are several sources of microorganisms: groundwater, surface water, and the different biofilms. Our objective was to estimate the proportion of microbial communities from surficial water or biofilm layers (sources) contributing to the formation of the microbial communities in deeper water or biofilm layers (sinks) or inversely. Our FEAST analysis confirmed a vertical microbial colonization, as surficial communities were sources for deeper layers (sinks) and vice versa. The fluxes of planktonic communities observed into the mine, confirm a connectivity between the surface water and groundwater microbial communities. We also observed a contribution of planktonic populations to the overall sessile community and from sessile population to planktonic communities. The transition from a planktonic to a sessile lifestyle confers many advantages to biofilm members, including protection when environmental conditions are hostile [98]. In contrast, the dispersion of cells from biofilms (sessile to planktonic communities) constitutes an essential part of microbial dissemination and colonization of new substrates [77,99]. Overall, vertical fluxes of archaeal populations seemed to be more dynamic than bacterial and eukaryote populations, possibly explained by their better adaptation or tolerance to hostile environments. However, unknown microbial communities still represent an important proportion of potential sources. Other potential sources might come from soil, plants, rainwater, or snow, and we need further studies to reveal the different origins of these communities to understand their complex structure. At 91 m deep, the three domains had very low unknows sources, suggesting the enrichment of deep communities by a downward flow of communities. Moreover, at 12 m deep, there was no or a very low contribution of communities from the 91 m layer, suggesting a limit of the upward groundwater flux.

Future work will need to focus on characterizing metallic trace metals concentration, especially the bioaccumulation in planktonic microorganisms and in biofilms for better understanding of interactions between biofilm metal content, elements in the water column, and the composition and structure of the mine microbiome.

### 5. Conclusions

Contrary to our initial assumption based on previous studies, depth was not a significant environmental factor explaining diversity and community composition in the Forsyth iron mine. However, microbial communities seem to disperse in the mine since surface communities contributed to the deeper ones and vice versa. We observed a vast diversity of microorganisms in all three domains (Bacteria, Archaea, and Eukaryote) with metabolisms that seem to be centered on carbon- and sulfur-cycling, although these observations need to be supported by future metagenomic analyses. We determined that lifestyle (sessile/planktonic) was the major environmental parameter explaining variance in all three domains. Taxa found in the sessile communities were based on $C_1$-compound cycling

in the deeper layers of the mine with a likely adaptation to heavy metal stress that would accumulate in the biofilms. The planktonic community was dominated by ultra-small archaea and bacteria taxa, as well as probable eukaryotic predators of prokaryotes, indicating immensely stressful conditions in this oligotrophic aquatic habitat. Many sequences were affiliated with unknown organisms with a potential for biodegradation of anthropogenic pollutants like heavy metal or hydrocarbons.

**Supplementary Materials:** The following supporting information can be downloaded at: https://www.mdpi.com/article/10.3390/applmicrobiol3010004/s1, Figure S1: Physicochemical profile of mine water along the depth gradient, $SO_4^{2-}$ (a), $Cl^-$ (b), $F^-$ (c), DIC (d), DOC (e), $n = 2$.; Table S1: PCR conditions for the amplification of the 16S/18S rRNA genes.; Figure S2. Isotopic composition of Forsyth mine water ($n = 21$). The Global meteoric water line (GMWL) and Canadian meteoric water line (CMWL) derived from the data are also shown. Isotopic values are reported in permil units (‰) against the Vienna Standard Mean Oc. Values are depth in meters.; Table S2. Spearman correlation between depth and all physicochemical parameters ($n = 2$); Figure S3. Shannon diversity indices (a), and ASV richness (b) for the three domains (mean $\pm$ std, $n = 37$, $n = 34$, $n = 40$, respectively); Table S3. Spearman correlation between alpha diversity indices and depth or all physicochemical parameters, for bacterial, archaeal, eukaryote communities.; Figure S4. Boxplots showing Shannon index differences between lifestyles for bacteria (a), archaea (b) and eukaryote (c) communities, $n = 37$, $n = 34$, $n = 40$, respectively. Significant differences $p < 0.05$.; Table S4. Permutation homogeneity tests between lifestyle, substrate type and depth gradient.; Figure S5. Boxplots showing ASV richness differences between lifestyles for bacteria (a), archaea (b) and eukaryote (c) communities, $n = 37$, $n = 34$, $n = 40$, respectively. Significant differences $p < 0.05$.; Figure S6. Relative abundance (% of total reads) for bacteria (a), phyla (b) and eukaryote (c) communities, at the genus level.

**Author Contributions:** Conceptualization, C.S.L.; methodology, E.L., F.C. and C.S.L.; validation, E.L. and C.S.L.; formal analysis, E.L.; investigation, E.L., F.C., K.B. and A.D.; resources, C.S.L.; data curation, E.L.; writing—original draft preparation, E.L.; writing—review and editing, C.S.L. and M.R.; visualization, E.L.; supervision, C.S.L. and M.R.; project administration, C.S.L., D.J. and V.P.; funding acquisition, C.S.L., D.J. and V.P. All authors have read and agreed to the published version of the manuscript.

**Funding:** This research was supported by the Canada Research Chair in "Aquatic Environmental Genomics", and a Natural Sciences and Engineering Research Council (NSERC) discovery grant [RGPIN-2019-06670] both awarded to CSL, as well as the NSERC and Fonds de Recherche—Nature et Technologie (FRQNT) [2023-NOVA-314112] grant awarded to C.S.L., V.P. and D.J. We thank the Interuniversity Research Group in Limnology (Groupe de Recherche Interuniversitaire en Limnologie—GRIL) and their funders, the FRQNT (Québec).

**Data Availability Statement:** The raw reads have been deposited into the National Center for Biotechnology Information (NCBI) under the PRJNA916235 accession number.

**Acknowledgments:** The authors would like to thank the PTO Exploration diving team, David Caissy, Gabriel Benoit-Martin, Rob deProy, Steve Doyon, for their incredible and crucial participation in this project, and for their invaluable help in collecting the samples. We would also like to thank the National Capital Commission (NCC) for granting us access to the mine.

**Conflicts of Interest:** The authors declare no conflict of interest.

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
