# Peer review of "Bacterial, Archaeal, and Eukaryote Diversity in Planktonic and Sessile Communities Inside an Abandoned and Flooded Iron Mine (Quebec, Canada)"

_2673-8007, doi:10.3390/applmicrobiol3010004_

Round 1

Reviewer 1 Report

Dear Authors,

Thank you for your manuscript submission “Bacterial, archaeal, and eukaryote diversity in planktonic and 2 sessile communities inside an abandoned and flooded iron mine 3 (Quebec, Canada).” The article studied the abundance and structure of microbial communities and the effects of some factors on their diversity in water and biofilm samples from an old iron mine in Quebec. Here are my specific comments:

-          More quantitative results should be added to the abstract

-          Lines 51-54, reference(s) is required

-          Line 98, samples were collected in 08 and 09/2019. Do the microbial communities change after three years and in different seasons/months?

-          Figure 1, the quality of this figure is low and hard to see the content. Replacing a new figure should be considered

-          The basic parameters of water samples, such as pH or turbidity, should be provided

-          Line 153 and line 253, typo “Error! Reference source not found”

-          Are any equations used for the calculation and statistical analysis? Should any assumptions be made?

-          Are there any limitations for the method in this study?

-          Future research is missing

-          The format of some references, such as ref 2, 3, 23, etc. does not follow journal guideline and need double-checking

Author Response

Reviewer 1

We would like to thank the reviewer for taking the time to review our study during the end-of-year holidays, and in such a short time frame. We would also like to thank the reviewer for her/his comments and helping to improve the manuscript.

More quantitative results should be added to the abstract

Reply from authors: We would argue here that our quantitative data is mostly percentage of relative abundance of taxa, and some geochemical data. We do not think it would strengthen the abstract to such numbers.

Lines 51-54, reference(s) is required

Reply from authors: Reference was added L55

Line 98, samples were collected in 08 and 09/2019. Do the microbial communities change after three years and in different seasons/months?

Reply from authors: It is to be expected that communities will change over time. We would have liked to continue this research with temporal and seasonal approaches but unfortunately, it was impossible for us to return to sample this mine, as we no longer have authorization access. But these approaches will be considered for the study of another mine. However, given the potential limited connection between surface water and the mine water, as well as the fact that the water inside the mine is always around 5°C, independent of the season, we do not expect time to have such a great impact on the observed results.

Figure 1, the quality of this figure is low and hard to see the content. Replacing a new figure should be considered

Reply from authors: This was done.

The basic parameters of water samples, such as pH or turbidity, should be provided

Reply from authors: The quantity of water that can be collected by divers is extremely limited since the water is cold and they need to wear protective gear, and they must carry their oxygen bottles. Also, the mine grounds belong to a national park, and we needed authorization from them to dive, and they granted us only 8 dives in total. For this reason, we chose not to sacrifice most of the water samples we were able to collect and kept them mainly for microbial analyses. Indeed, we did not want to dip the probe inside the bottles and risk contaminating them. For these reasons, some physico-chemical parameters of the water are missing for some of the dives and we were not able to use the data.

Line 153 and line 253, typo “Error! Reference source not found”

Reply from authors: This was added L153 and L253.

Are any equations used for the calculation and statistical analysis? Should any assumptions be made?

Reply from authors: No equations were used.

Are there any limitations for the method in this study?

Reply from authors: Most of the limitations occurred during sampling, especially the volume of water collected at each dive, and the possible water mixing.

Future research is missing

Reply from authors: Actually, the future work is explained L565-568.

The format of some references, such as ref 2, 3, 23, etc. does not follow journal guideline and need double-checking

Reply from authors: This has been modified.

Reviewer 2 Report

This article entitled,"Bacterial, archaeal, and eukaryote diversity in planktonic and sessile communities inside an abandoned and flooded iron mine (Quebec, Canada)" is a very well-written article and describes some interesting features regarding the microbial communities development and accumulation over the time. There are a couple of comments for improvement, the rest of the manuscript is good to go. I found reading this article very interesting and informative. the study is worth reading and replicating of course.

Author Response

We would like to thank the reviewer for taking the time to review our study during the end-of-year holidays, and in such a short time frame. We would also like to thank the reviewer for her/his comments and helping to improve the manuscript, and especially for the positive remarks. This is rare during the review process but helps as much as other comments.

I think this word eukaryotes should be replaced with fungi as this somehow misleading here as microeukaryotes also include some members from other domains like plantae as well

Reply from authors: We would argue here that we studied more than fungi in the mine, we mention other microeukaryotes in the discussion. For example, we mentioned Gastritricha and Colpodella L383, L453 and L457 and, Ciliophora and Arthropoda L518.

Section 2.3. needs a little bit of more elaboration on the DNA extraction and purification protocol if some researcher wants to replicate the process in case.

Reply from authors: We used exactly the manufacturer’s instructions included in the Qiagen extraction kit. There was no further purification of the DNA after using the kit.